# Identification and Quantification by NMR Spectroscopy of the 22*R* and 22*S* Epimers in Budesonide Pharmaceutical Forms

**DOI:** 10.3390/molecules27072262

**Published:** 2022-03-31

**Authors:** Natalia E. Kuz’mina, Sergey V. Moiseev, Elena Y. Severinova, Evgenii A. Stepanov, Natalia D. Bunyatyan

**Affiliations:** 1Scientific Centre for Expert Evaluation, Medicinal Products of the Ministry of Health of the Russian Federation, Federal State Budgetary Institution, 8/2 Petrovsky Blvd, 127051 Moscow, Russia; moiseevsv@expmed.ru (S.V.M.); severinova@expmed.ru (E.Y.S.); bunyatyan@expmed.ru (N.D.B.); 2Department of Chemistry and Chemistry Education, Charles University, Ovocný trh 560/5, 116 36 Prague, Czech Republic; orexrom@gmail.com; 3Department of Pharmaceutical Technology and Pharmacology, I.M. Sechenov First Moscow State Medical University (Sechenov University), 8, Bldg. 2 St. Trubetskaya, 119991 Moscow, Russia

**Keywords:** budesonide, 22*R* and 22*S* epimers, identification, quantification, qNMR, HPLC

## Abstract

The authors developed four variants of the qNMR technique (^1^H or ^13^C nucleus, DMSO-d6 or CDCl_3_ solvent) for identification and quantification by NMR of 22*R* and 22*S* epimers in budesonide active pharmaceutical ingredient and budesonide drugs (sprays, capsules, tablets). The choice of the qNMR technique version depends on the drug excipients. The correlation of ^1^H and ^13^C spectra signals to molecules of different budesonide epimers was carried out on the basis of a comprehensive analysis of experimental spectral NMR data (^1^H-^1^H gCOSY, ^1^H-^13^C gHSQC, ^1^H-^13^C gHMBC, ^1^H-^1^H ROESY). This technique makes it possible to identify budesonide epimers and determine their weight ratio directly, without constructing a calibration curve and using any standards. The results of measuring the 22*S* epimer content by qNMR are comparable with the results of measurements using the reference HPLC method.

## 1. Introduction

Budesonide [Bud; 22(*R,S*)-(11β,16α)-16,17-Butylidenebis(oxy)-11,21-dihydroxypregna-1,4-diene-3,20-dione] is a synthetic compound of the glucocorticoid family with anti-inflammatory, anti-allergic, and immunosuppressive effects. Bud is actively used for the topical treatment of asthma, rhinitis, and inflammatory bowel disease [1,2,3,4,5] and included in the WHO list of essential medicines.

Bud is a racemic mixture of two epimers (22*R* and 22*S*, Figure 1). The epimers ratio in the mixture is determined by the synthesis method [6]. Although they have similar qualitative pharmacological effects, the Bud-22*R* is several times more potent than Bud-22*S* [7,8]. Therefore, the content of the less active epimer in the Bud active pharmaceutical ingredient (API) and Bud drug products is strictly normalized.

The identification and quantification of the Bud-22*R* and Bud-22*S* are carried out by capillary gas chromatography [6], high performance liquid chromatography (HPLC) [9,10], and sensitive ultra-high-performance liquid chromatography–tandem mass spectrometry method (HPLC-MS) [11,12]. These methods identify Bud epimers indirectly by comparing test samples with reference standards. Quantitative measurements by GC, HPLC, and HPLC-MS methods are relative and include the step of building a calibration function using a reference standard of the measured compound. It is important to use absolute and direct methods to identify and quantify Bud epimers. Absolute and direct methods (for example, qNMR) do not require the use of reference standards and the construction of calibration functions. The aim of this article is to develop the technique of the identification and quantification using qNMR of Bud-22*R* and Bud-22*S* in APIs and Bud drugs. The developed technique will allow selective identification of Bud-epimers and quantitative evaluation of its weight ratio directly by recording the characteristic signals of Bud-22*R* and Bud-22*S* in the NMR spectra and measuring their integral intensities. 

## 2. Results and Discussion

The simplest option for structural interpretation is the Bud-API spectrum, since it does not contain excipient signals. The comprehensive analysis of spectral data from 2D experiments (^1^H-^1^H gCOSY, ^1^H-^13^C gHSQC, ^1^H-^13^C gHMBC, ^1^H-^1^H ROESY) allowed us to correlate the ^1^H and ^13^C signals to different epimer molecules (Table 1). 

The C22-H bond direction (*S* or *R*) in each of the two epimers was determined by the technique ^1^H-^1^H ROESY. Only Bud-22*R* has protons C16-H and C22-H on the same side of the 1,3-dioxolane ring (Figure 1). This is the reason for the appearance of cross-peaks between these valence unbound protons in the ROESY spectrum. Figure 2 shows a fragment of the ROESY spectrum of Bud-API in DMSO-d6, containing the C16-H and C22-H proton signals (δ 4.75 and 4.52 ppm for one epimer and 5.05 and 5.17 ppm for the other). Only the proton pair 4.75–4.52 ppm had cross-peaks. This fact indicates that protons 4.75 and 4.52 ppm belong to the Bud-22*R*. The proton pair 5.05–5.17 ppm is part of the Bud-22*S*. 

It should be noted that the Bud NMR spectral data presented in the literature [13,14] lack structural correlation of Bud NMR spectra signals to specific 22*R* and 22*S* epimers. 

The spectra analysis of Bud-API solutions in DMSO-d6 and CDCl_3_ (Figure 3, Figure 4, Figure 5 and Figure 6, Table 1) allowed to determine isolate signals for each epimer (characteristic signals). There are following characteristic signals for Bud-22*R*: 

(1) ^1^H (DMSO-d6), δ, ppm: 4.13 d (C21-H), 4.39 d (C21-H), 4.52 t (C22-H); 

(2) ^1^H (CDCl_3_), δ, ppm: 4.24 d (C21-H), 4.54 t (C22-H), 4.89 d (C16-H);

(3) ^13^C (DMSO-d6), δ, ppm: 66.00 (C21), 80.83 (C16), 97.17(C17); 103.42 (C22);

(4) ^13^C (CDCl_3_), δ, ppm: 46.09 (C13); 49.90 (C14), 82.26 (C16), 97.31 (C17), 104.80 (C22).

There are the following characteristic signals for Bud-22*S*:

(1) ^1^H (DMSO-d6), δ, ppm: 4.06 d (C21-H), 4.45 d (C21-H), 5.05 d (C16-H), 5.17 t (C22-H); 

(2) ^1^H (CDCl_3_), δ, ppm: 4.19 d (C21-H), 4.61 d (C21-H);

(3) ^13^C (DMSO-d6), δ, ppm: 65.60 (C21), 81.90 (C16), 97.92(C17); 107.04 (C22);

(4) ^13^C (CDCl_3_), δ, ppm: 47.51 (C13); 52.92 (C14), 83.52 (C16), 97.99 (C17), 108.54 (C22).

It should be noted that the use of DMSO-d6 provides a better separation of the characteristic signals of the Bud-22*R* and Bud-22*S* epimers in the proton spectrum. CDCl_3_ provides better ^13^C spectrum resolution.

The characteristic signals can be spectral markers of these epimers in the analyzed sample. Their normalized integral intensities are equal to the fraction of each epimer in the racemate mixture. It should be noted that qNMR is considered in the literature as an absolute and direct method for measuring the molar ratio of the analytes in a test sample, as well as the weight content of one component relative to another component, because the functional relationships between the analytes and the measurands (integrated intensities) are well-known: the molar ratio of the components in a mixture is equal to the ratio of the normalized integrated intensities of the signals of these components. Uncertainty of the measuring result by qNMR relies only on the uncertainty of the integral intensities ratio measurement [15]. The results of measurements by HPLC (pharmacopeial method) are relative and indirect by nature. Determination of Bud-22*R* and Bud-22*S* epimers by HPLC requires generation of a calibration curve using their pharmacopeial reference standards (the relative nature of measurements). The measurement by the HPLC method has a combined uncertainty (the indirect nature of measurements). Sources of the total uncertainty are the peak area measurement in the chromatogram, weighing of the test and standard samples, and solvent volume measurements. Therefore, the accuracy of measurement of Bud epimeric composition by direct and absolute method qNMR is higher than by indirect and relative method HPLC. Moreover, both normalized integral intensities of a selected individual pair of 22*R* and 22*S* epimeric signals and the average value of pairwise normalized integral intensities of all observed pairs of characteristic signals can be taken as a result of measuring the epimeric composition of the Bud sample. Averaging the measurement results reduces its uncertainty. In chromatographic methods, averaging is only possible with a series of experiments.

Bud drugs of different manufacturers have in their content a nonequal set of excipitents. The solubility of excipients influences the choice of solvent (DMSO-d6 or CDCl_3_) For example, a nasal spray is an aqueous suspension of Bud. The excipitents of this suspension are DMSO-soluble sodium methylparaben, carboxymethylcellulose and sodium carmellose, polysorbate 80, sucrose, polypropylene glycol and disodium edetate. Obviously, it is appropriate to use CDCl_3_ rather than DMSO-d6 when analyzing this drug. The sample extraction with chloroform will concentrate Bud and remove excipients that do not pass into the extractant. In the ^1^H (CDCl_3_) spectrum of the Bud nasal spray, all characteristic signals of the Bud-22*R* and Bud-22*S* are observed (Figure 7a). For quantitative measurements, it is reasonable to use the most isolated signals 4.89 d (22*R*) and 4.61 d (22*S*). In the ^13^C (CDCl_3_) spectrum of this preparation, all characteristic signals are also present (Figure 7b). 

Bud capsules contain chloroform-insoluble lactose monohydrate; therefore, it is also advisable to use CDCl_3_ for this drug. The characteristic signals ^1^H and ^13^C of the Bud-22*R* and Bud-22*S* for capsule solutions in CDCl_3_ are shown in Figure 8.

Bud tablets contain excipients with different solubility in DMSO-d6 and CDCl_3_: stearic acid, soy lecithin, cellulose, hydroxypropylcellulose, lactose monohydrate, and magnesium stearate. The characteristic signals of Bud epimers partially overlap with the signals of excipients in ^1^H spectra of Bud tablets solution in DMSO-d6 and CDCl_3_ (Figure 9). For this reason, precise quantitative measurements are not possible. When selecting the ^3^C nucleus, isolated characteristic signals are observed for each solvent (DMSO-d6 and CDCl_3_; Figure 10). 

Table 2 shows the results of quantitative measurements of the 22*S* and 22*R* epimers content in the Bud-API and Bud drugs, obtained using different versions of the developed technique. 

In the ^13^C spectra, characteristic signals of Bud-22*R*, Bud-22*S*, and excipients are located at a considerable distance from each other. Therefore, the signal ^13^C can be integrated using the general rule for choosing the integration limit (the integration limit is equal to 64 times the half-width of a Lorentzian shape NMR signal [15]). In the ^1^H spectra of Bud drugs, there is a partial overlap of the signals of Bud and excipients in this frequency range. Therefore, the integration limit of the Bud epimer signals in the ^1^H spectra were narrowed to 20 times the half-width of a Lorentzian shape. It should be noted that variation in the solvent and nucleus does not affect the result of quantitative measurement of the Bud epimers content. For example, the RSD of the measurement results of Bud-22*S* content in Bud-API is 0.15% (mean volume is 47.52%). 

The results of measurement of 22*S* epimer content using qNMR are comparable with the results of the HPLC reference method. Thus, the content of Bud-22*S* in the API and nasal spray, measured by HPLC, was 47.3 and 46.8% (47.52 and 46.60% by qNMR). The similarity of the measurement results, obtained by qNMR and HPLC methods, is an additional proof of the correctness of the proposed technique.

## 3. Materials and Methods

### 3.1. Materials 

The following materials were used in the qNMR technique development: Bud-API by Farmabios S.p.A., Italy (A), nasal spray «Tafen Nasal» by Lek d.d., Slovenia (B), Bud capsules «Respinid» by Sava Healthcare Limited, India (C), tablets «Kortiment» by Cosmo S.p.A., Italy (D). Deuterated dimethylsulfoxide (DMSO-d6, 99.90% D) and chloroform (99.8% D) by Cambridge Isotope Laboratories, Inc. (St. Louis, MO, USA) were used in the NMR experiments. 

HPLC measurements were carried out using the certified reference standard for Bud, manufactured by the European Pharmacopoeia, glacial acetic acid, potassium hydroxide (Sigma-Aldrich, Saint Louis, MO, USA). HPLC grade acetonitrile was purchased from Fisher Scientific (Fairlawn, NJ, USA). HPLC ready 18 MΩ water was obtained, in-house, from a Milli-Q Integral 3 water purification system, Merck Millipore Corp. (Burlington, MA, USA). Duran filter funnels (porosity 3) were used for filtration.

### 3.2. NMR Spectroscopy Method

#### 3.2.1. Sample Preparation

*API*: About 20 mg of the Bud-API (exact mount is optional) were placed in an NMR tube, 0.5 mL of solvent (DMSO-d6 or CDCl_3_) was added, shaken vigorously until the sample was completely dissolved.

*Nasal spray*: The contents of 1 vial was transferred to a separating funnel, 2 mL of CDCl_3_ were added and thoroughly shaken for 5 min; then, the bottom organic layer was separated and transferred to the NMR tube.

*Capsules*: 10 mL of CHCl_3_ were added to the contents of 30 capsules, thoroughly mixed and filtered; then, the filtrate was centrifuged. The supernatant was separated and dried by air. The resulting dry residue was dissolved in 0.6 mL CDCl_3_ and transferred to an NMR tube.

*Tablets DMSO-d6*: 3 mL of DMSO-d6 were added to the two powdered tablets, thoroughly mixed and filtered; then, the filtrate was centrifuged. A total of 0.6 mL of the supernatant was separated and transferred to the NMR tube.

*Tablets, CDCl_3_*: 10 mL of CHCl_3_ were added to the 2 tablets crushed into a powder, thoroughly mixed and filtered; then, the filtrate was centrifuged. The supernatant was separated and dried by air. The resulting dry residue was dissolved in 0.6 mL of CDCl_3_ and transferred to an NMR tube.

#### 3.2.2. Instrumentation and Experiment Conditions 

NMR spectra were collected on the Agilent DD2 NMR System 600 NMR spectrometer equipped with a 5 mm broadband probe and a gradient coil (VNMRJ 4.2 software). Parameters of 1D experiments: temperature—27 °C; spectral width—6009.6 Hz (^1^H) and 37,878.8 Hz (^13^C); observed pulse 90° (^1^H) and 45° (^13^C); acquisition time—5.325 s (^1^H) and 0.865 s (^13^C); relaxation delay—10 s (^1^H) and 1 s (^13^C); number of scans—256 (^1^H) and 10,000 (^13^C); the number of analog-to-digital conversion points—64 K; exponential multiplication—0.3 Hz (^1^H) and 3 Hz (^13^C); zero filling—64 K; automatic linear correction of the spectrum baseline, manual phase adjustment, calibration of the δ scale under DMSO (δ = 2.50 ppm for ^1^H and 39.52 ppm for ^13^C) or CHCl_3_ (δ = 7.26 ppm for ^1^H and 77.16 ppm for ^13^C) [16]. The manual mode was also used for the signal integration. The integration limit was equal to 20 (^1^H) and 64 (^13^C) times the half-width of a Lorentzian shape NMR signal. The relaxation delay value was estimated using an inversion-recovery experiment: T1 is equal to 1.55 s. The ROESY experimental parameters: the relaxation time—1 s; the number of free induction signal accumulation per increment—16; the number of analog-to-digital conversion points—2K × 256; the mixing time—0.2 s; the pulse duration—0.15 s. 

### 3.3. Reference Measurement with HPLC Method

#### 3.3.1. Preparation of Solution

System suitability test solution, buffer solution, test solution of samples A–D, reference solutions, and mobile phase were prepared according to USP methods [9,10]. 

#### 3.3.2. Instrumentation and Chromatographic Conditions

The HPLC system consists of an Agilent Infinity 1260 series (Agilent Technologies, Wilmington, DE). Data collection and analysis were performed using ChemStation software. Chromatographic conditions: column—Zorbax RX-C-18 250 mm × 4.6 mm × 5 µm (Agilent Technologies, Santa-Clara, CA, USA); column temperature—50 °C; mobile phase—acetonitrile and buffer pH 3.9 (45:55) for sample A and acetonitrile and water (70:30) for sample B; flow rate—1 mL/min; detector—UV 240 nm for sample A and 245 nm for sample B; injection volume—20 μL for sample A and 50 μL for sample B; run time—no less 40 min.

## 4. Conclusions

Different versions of the qNMR technique for identification and quantification Bud-22*R* and Bud-22*S* epimers (^1^H or ^13^C core, DMSO-d6 or CDCl_3_ solvent) were developed for Bud APIs and Bud drugs. This technique does not need Bud-epimers reference standards. The choice of the qNMR technique version depends on the drug excipients in Bud drugs. Application of this technique will reduce the uncertainty of the measurement result, since the experimental procedure does not contain the stages of taking accurate weights, volumes, and constructing a calibration curve. This technique can be used for carrying out GP APIs and drug analyses.

## Figures and Tables

**Figure 1 molecules-27-02262-f001:**
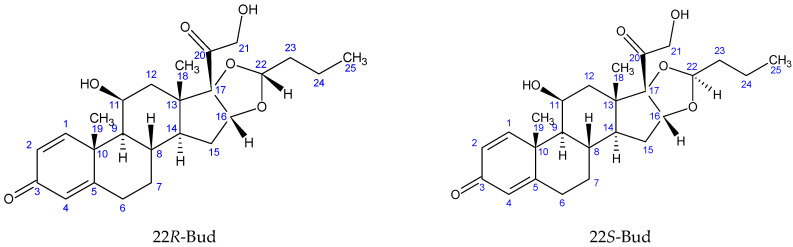
Chemical structures of Bud-22*R* and Bud-22*S*.

**Figure 2 molecules-27-02262-f002:**
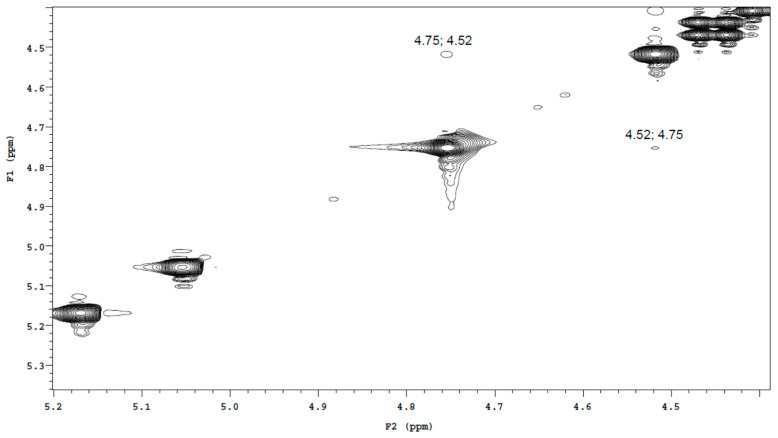
^1^H-^1^H ROESY spectrum fragment of the Bud-API solution in DMSO-d6.

**Figure 3 molecules-27-02262-f003:**
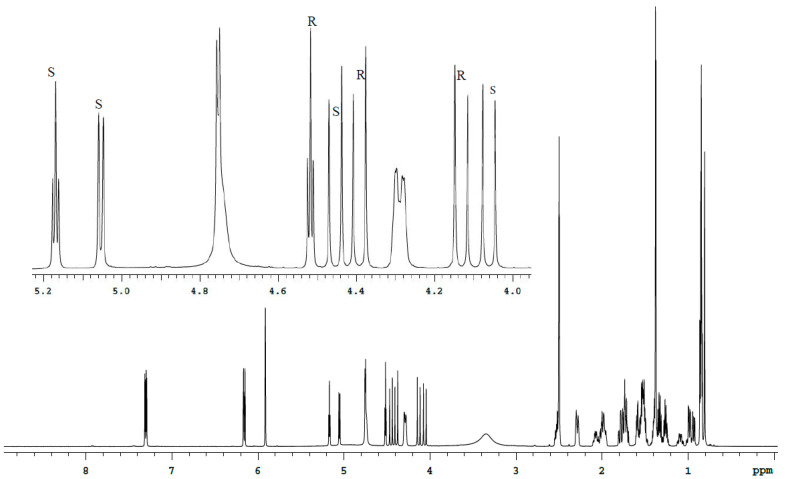
^1^H spectrum of the Bud-API solution in DMSO-d6.

**Figure 4 molecules-27-02262-f004:**
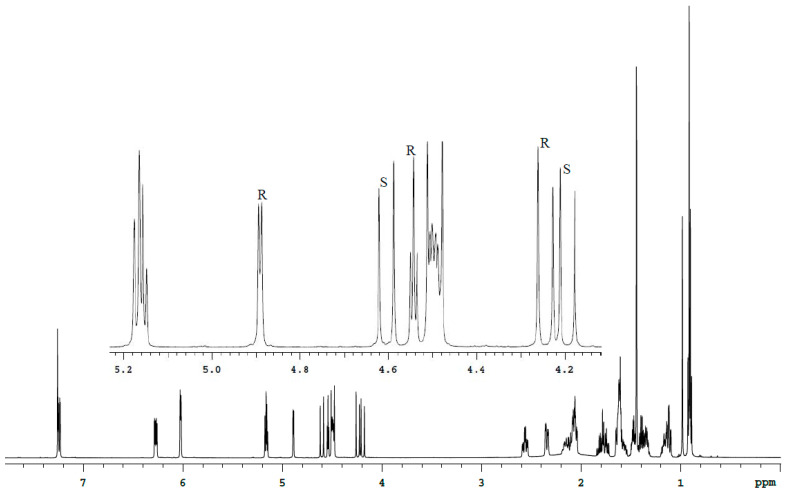
^1^H spectrum of the Bud-API solution in CDCl_3_.

**Figure 5 molecules-27-02262-f005:**
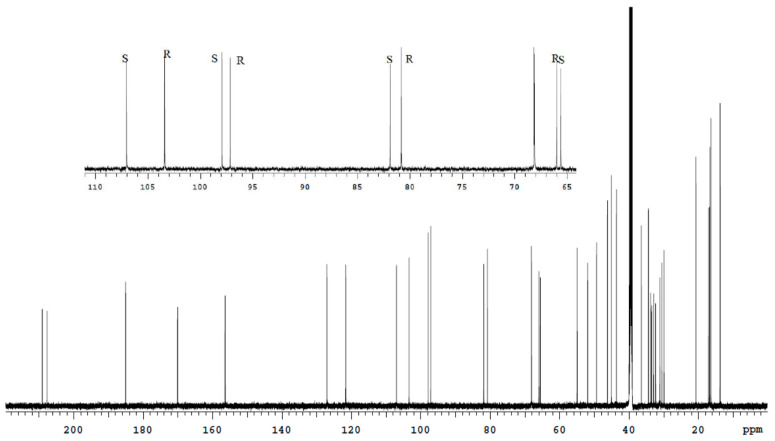
^13^C spectrum of the Bud-API solution in DMSO-d6.

**Figure 6 molecules-27-02262-f006:**
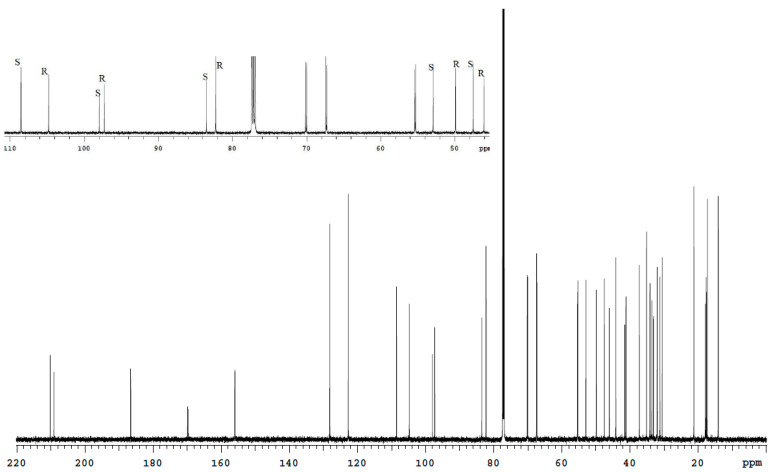
^13^C spectrum of the Bud-API solution in CDCl_3_.

**Figure 7 molecules-27-02262-f007:**
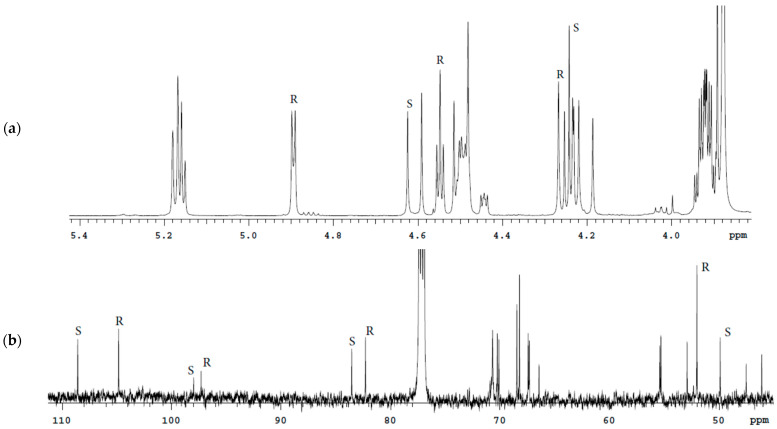
^1^H (**a**) and ^13^C (**b**) spectra fragments of the Bud nasal spray solution in CDCl_3_ with characteristic signals of 22*R* and 22*S* epimers.

**Figure 8 molecules-27-02262-f008:**
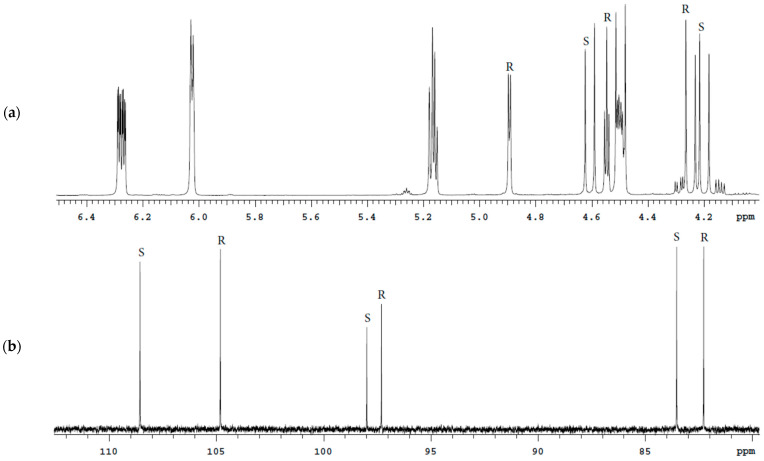
^1^H (**a**) and ^13^C (**b**) spectra fragments of the Bud capsules solution in CDCl_3_ with characteristic signals of 22*R* and 22*S* epimers.

**Figure 9 molecules-27-02262-f009:**
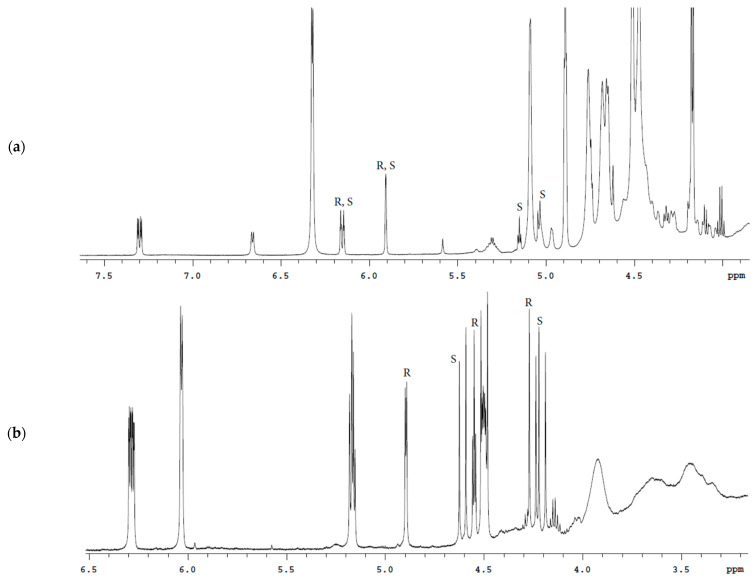
^1^H spectra fragments of the Bud tablets solutions in DMSO-d6 (**a**) and CDCl_3_ (**b**) with characteristic signals of 22*R* and 22*S* epimers.

**Figure 10 molecules-27-02262-f010:**
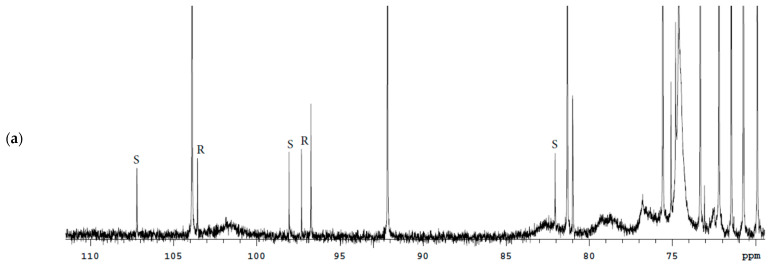
^13^C spectra fragments of the Bud tablets solutions in DMSO-d6 (**a**) and CDCl_3_ (**b**) with characteristic signals of 22*R* and 22*S* epimers.

**Table 1 molecules-27-02262-t001:** Spectral characteristics of 22*R*-Bud and 22*S*-Bud.

No.	22*R*		22*S*	
	δ, ppm		δ, ppm	
	^1^H	^13^C	^1^H	^13^C
DMSO-d6				
1	7.31 d (J = 10.0)	156.40	7.30 d (J = 10.0)	156.43
2	6.16 dd (J = 10.0; 1.9)	127.11	6.16 d (J = 10.0; 1.9)	127.08
3		185.08		185.06
4	5.91 br.s	121.67	5.91 br.s	121.62
5		170.09		170.16
6	2.29 m; 2.52 m	31.17	2.29 m; 2.52 m	31.15
7	1.07 dd (J = 12.3; 4.7); 2.00 m	33.84	1.11 dd (J = 12.3; 4.7); 1.96 m	33.51
8	2.07 m	29.97	2.01 m	30.58
9	0.99 dd (J = 11.2; 3.5)	55.01	0.94 dd (J = 11.2; 3.5)	54.99
10		43.64		43.66
11	4.30 m	68.17	4.28 m	68.13
12	1.73 m	39.34	1.78 m	39.57
13		45.14		46.26
14	1.51 m	49.39	1.52 m	51.96
15	1.52 m; 1.59 m	32.93	1.58 m; 1.72 m	32.38
16	4.75 d (J = 4.3)	80.83	5.05 d (J = 7.3)	81.90
17		97.17		97.92
18	0.81 s	16.84	0.85 s	17.50
19	1.38 s	20.76	1.37 s	20.74
20		209.11		207.71
21	4.13 d (J = 19.4); 4.39 d (J = 19.4)	66.00	4.06 d (J = 19.2); 4.45 d (J = 19.2)	65.60
22	4.52 t (J = 4.5)	103.42	5.17 t (J = 4.8)	107.04
23	1.53 m	34.46	1.39 m	36.50
24	1.33 m	16.42	1.26 m	16.75
25	0.85 t (J = 7.4)	13.79	0.85 t (J = 7.4)	13.79
11-OH	4.74 br.s		4.74 br.s	
CDCl_3_				
1	7.25 d (J = 10.1)	156.01	7.24 d (J = 10.1)	156.04
2	6.28 dd (J = 10.1; 1.8)	128.14	6.27 dd (J = 10.1; 1.8)	128.14
3		186.63		186.58
4	6.03 br.s	122.71	6.02 br.s	122.71
5		169.88		169.75
6	2.35 ddd (J = 13.7; 4.5; 1.8) 2.56 ddd (J = 13.7; 13.5; 5.5)	32.02	2.35 ddd (J = 13.7; 4.5; 1.8) 2.56 ddd (J = 13.7; 13.5; 5.5)	32.00
7	1.17 m; 2.07 m	34.14	1.17 m; 2.07 m	34.11
8	2.16 m	30.54	2.11 m	31.19
9	1.12 m	55.31	1.12 m	55.41
10		44.14		44.14
11	4.50 br.d (J = 3.3)	70.16	4.49 br.d (J = 3.3)	70.08
12	1.63 m; 2.07 m	41.17	1.63 m; 2.07 m	41.51
13		46.09		47.51
14	1.61 m	49.90	1.57 m	52.92
15	1.61 m; 1.78 m	33.58	1.75 m; 1.82 m	33.13
16	4.90 d (J = 4.7)	82.26	5.17 d (J = 6.8)	83.53
17		97.31		97.99
18	0.92 s	17.56	0.98 s	17.85
19	1.44 s	21.23	1.45 s	21.22
20		210.26		209.17
21	4.24 d (J = 19.8); 4.50 d (J = 19.8)	67.41	4.19 d (J = 19.8); 4.61 d (J = 19.8)	67.31
22	4.54 t (J = 4.5)	104.80	5.16 t (J = 5.1)	108.54
23	1.62 m	35.13	1.48 m	37.22
24	1.39 m	17.25	1.35 m	17.56
25	0.92 t (J = 7.5)	14.09	0.90 t (J = 7.5)	14.06

**Table 2 molecules-27-02262-t002:** The quantitative measurements results of the content of 22*S* and 22*R* epimers in the Bud-API and Bud drugs.

Bud	Content of 22*S* (22*R*), %				
	DMSO-d6		CDCl_3_		Mean Volume
	^1^H	^13^C	^1^H	^13^C	
API	47.45 (52.55)	47.60 (52.40)	47.47 (52.53)	47.56 (52.44)	47.52 (52.48)
Nasal spray	-	-	46.67 (53.33)	46.53 (53.47)	46.60 (53.40)
Capsules	-	-	47.82 (52.18)	47.67 (52.33)	47.75 (52.25)
Tablets	-	48.60 (51.40)	-	48.81 (51.19)	48.71 (51.29)

## Data Availability

The data that support the findings of this study are available from the corresponding author upon reasonable request.

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
