# Peer review of "Identification and Quantification by NMR Spectroscopy of the 22R and 22S Epimers in Budesonide Pharmaceutical Forms"

_molecules, 2022, doi:10.3390/molecules27072262_

Round 1

Reviewer 1 Report

The authors of the manuscript have tried to work out a simple tool for analyzing the ratio of 22R and 22S epimers of Budesonide in various pharmaceutical forms. The method is based on the combination of 1H and 13C NMR experiments in two solvents, i.e. DMSO and CDCl3. The have managed to assign all resonances to both forms and to choose the characteristic signals that discriminate 22R and 22S epimers. The method can be used even for quantitative analysis in various pharmaceutical forms, like nasal spray, tablets or capsules.

 The manuscript is clearly written, all conclusions are supported by experimental results. English language is decent, I have found just several typographical errors, the list of which follows:

l. 99: nonequal set – a nonequal set

l. 118: there is incorrect formatting 13C – 13C

l. 166 and further: NMR ampoule - NMR tube

l. 169: mL – ml

l. 173-174: the sentence: “The supernatant was separated and dried in a current of air.” Should be corrected to “The supernatant was separated and dried by air.”

l. 177: is – was (2x)

l. 177: ml is missing, just behind 0.6

l. 188: the waiting period between scans is 10s for 1H, but only 1s in case 13C. Is it correct?

Author Response

Dear reviewer!

We are grateful for the in-depth analysis of the article and the noted remarks. All typos you noted have been corrected.

Using different waiting periods between scans for 1H and  13C is correct, since only signals of the same-type C atoms with the same longitudinal relaxation time T1 are integrated in the 13C spectrum. In the 1H spectrum, the signals of nuclei with different relaxation times are integrated, so for 1H we calculated the waiting period between scans taking into account the maximum value of T1

Reviewer 2 Report

This article provides a NMR method to identify and quantify the 22R and 22S epimers in budesonide forms. The experimental design and conclusion of the article are not a problem, and it is reasonable to use NMR to identify the epimers. However, I'm not quite sure that the article is sufficiently innovative and its application seems too narrow. Here I point out some small suggestions:

  1. Mark the main difference signals in the NMR spectrum, which is convenient for readers to read.
  2. The format of DMSO-d6 is not standardized.

Author Response

Dear reviewer!

Thank you for your analysis of our article. Your comments are taken into account. The characteristic signals of the epimers 22R and 22S are marked on the spectra of budesonide drugs.

The abbreviation DMSO  is a common one. For example: Gottlieb, H.E.; Kotlyar, V. NMR Chemical Shifts of Common Laboratory Solvents as Trace Impurities. J. Org. Chem. 1997, 62, 7512–7515. The abbreviation DMSO is  deciphered in section 3.1. Materials.

We would like to draw your attention to the fact that the spectral characteristics of the 22S epimer are not presented in the literature and there is no structural correlation of NMR signals (1H or 13C) of the racemic mixture of 22R and 22S budesonide epimers. In addition, no technique for determining the epimeric composition of budesonide in tablets and capsules by any method are presented in the literature. This information is presented in our article, which makes it quite innovative. Many countries produce budesonide in different dosage forms, so the technique for assessing its quality is relevant to both manufacturers and regulatory agencies.

Reviewer 3 Report

Comment to the author:

The authors have developed the four variants of techniques (1H or 13C nucleus, DMSO-d6 or CDCl3 solvent) of the identification and quantification by qNMR of Bud-22R and Bud-22S in APIs and Bud drugs. The qualitative and quantitative techniques of epimers are expounded and the advantages and disadvantages of various techniques for different drugs excipients were summarized. The article is available from pharmaceutical analysis point of view and theme of the article meets the scope of the journal. After minor revisions, the article could be considered for publication. The following points should be noted:

  • In the discussion moiety, the author should tell the readers why application of this technique will reduce the uncertainty of the measurement result and is best designed for experimental proof (No experiment if it can be explained).
  • There are also some typos and grammatical errors need to be corrected. Please check it carefully.
  1. the “и” on line 47.
  2. the “nucleum” on line 118.

Author Response

Thank you for your analysis of our article. Your comments are taken into account. We added a fragment in the section “Results and discussion”, that explains to the readers why the application of this methodology will reduce the uncertainty of the measurement result. We have corrected all the typos and inaccuracies you noted.

Round 2

Reviewer 2 Report

The authors have revised the manuscript according to the reviewers suggestion. The manuscript can be acceppted.